# Objective Quantification of In-Hospital Patient Mobilization after Cardiac Surgery Using Accelerometers: Selection, Use, and Analysis

**DOI:** 10.3390/s21061979

**Published:** 2021-03-11

**Authors:** Frank R. Halfwerk, Jeroen H. L. van Haaren, Randy Klaassen, Robby W. van Delden, Peter H. Veltink, Jan G. Grandjean

**Affiliations:** 1Thoraxcentrum Twente, Medisch Spectrum Twente, P.O. Box 50 000, 7500 KA Enschede, The Netherlands; j.vanhaaren@mst.nl (J.H.L.v.H.); j.grandjean@mst.nl (J.G.G.); 2Department of Biomechanical Engineering, TechMed Centre, University of Twente, P.O. Box 217, 7500 AE Enschede, The Netherlands; 3Human Media Interaction Lab, University of Twente, P.O. Box 217, 7500 AE Enschede, The Netherlands; r.klaassen@utwente.nl (R.K.); r.w.vandelden@utwente.nl (R.W.v.D.); 4Department of Biomedical Signals and Systems, Faculty of Electrical Engineering, Mathematics and Computer Science, University of Twente, P.O. Box 217, 7500 AE Enschede, The Netherlands; p.h.veltink@utwente.nl

**Keywords:** wearable technology, early ambulation, thoracic surgery, activity classification, k-fold cross validation, LOO cross-validation, biomedical signal processing, patient monitoring

## Abstract

Cardiac surgery patients infrequently mobilize during their hospital stay. It is unclear for patients why mobilization is important, and exact progress of mobilization activities is not available. The aim of this study was to select and evaluate accelerometers for objective qualification of in-hospital mobilization after cardiac surgery. Six static and dynamic patient activities were defined to measure patient mobilization during the postoperative hospital stay. Device requirements were formulated, and the available devices reviewed. A triaxial accelerometer (AX3, Axivity) was selected for a clinical pilot in a heart surgery ward and placed on both the upper arm and upper leg. An artificial neural network algorithm was applied to classify lying in bed, sitting in a chair, standing, walking, cycling on an exercise bike, and walking the stairs. The primary endpoint was the daily amount of each activity performed between 7 a.m. and 11 p.m. The secondary endpoints were length of intensive care unit stay and surgical ward stay. A subgroup analysis for male and female patients was planned. In total, 29 patients were classified after cardiac surgery with an intensive care unit stay of 1 (1 to 2) night and surgical ward stay of 5 (3 to 6) nights. Patients spent 41 (20 to 62) min less time in bed for each consecutive hospital day, as determined by a mixed-model analysis (*p* < 0.001). Standing, walking, and walking the stairs increased during the hospital stay. No differences between men (*n* = 22) and women (*n* = 7) were observed for all endpoints in this study. The approach presented in this study is applicable for measuring all six activities and for monitoring postoperative recovery of cardiac surgery patients. A next step is to provide feedback to patients and healthcare professionals, to speed up recovery.

## 1. Introduction

Cardiac surgery consists mainly of coronary artery bypass grafting (CABG) followed by valve surgery [1]. The hospital stay for these patients is 5 to 7 days after surgery. Currently, automated monitoring of the mobilization process in the period surrounding the surgery is sparsely applied in practice.

### In-Hospital Mobilization

For patients, the importance of postoperative mobilization is often unclear. Most patients stay in bed during their hospital stay because they overestimate the intensity of the physical activities [2]. Next, the hospital environment does not stimulate patients to be physically active [3], and patients acquire a loss in muscle strength and aerobic capacity [4].

Hospitalized patients spent up to 50% lying in bed during daytime, despite the use of enhanced recovery after surgery protocols [5]. Others found that only a minority of patients had their meal outside of bed or walked during their hospital admission [6]. Inactivity results in a prolonged hospital stay, increased cardiovascular mortality and morbidity, and more readmissions to the intensive care unit (ICU) and to the hospital after discharge [7].

In-hospital exercises include breathing techniques, transfer between bed and chair, and walking along the ward. In the U.S.A., almost 20% of patients after cardiac surgery received no physical therapy or inpatient cardiac rehabilitation, and when this is utilized, often less than 30% of the hospital days are covered [8], without weekend day services [9]. Although interventions are hard to compare, the early mobilization of patients might improve physiological functional capacity, might accelerate recovery, and might reduce hospital length of stay [10,11]. Male patients after orthopedic and cardiac surgery showed higher levels and a steeper increase in mobilization compared to female patients [12,13].

The success of hospital mobilization can be assessed with patient-reported [14,15] or professional [16,17,18] scoring, or wearable devices [12,13,15,19,20,21,22]. Often these measures are subjective, the wound area is near the intended sensor placement, or do not cover all mobilization activities during the hospital stay. Some sensors have been used for cardiac surgery patients [12,20,22].

Studies in a hospital setting can be done with existing combinations of embedded sensors with proprietary algorithms but often depend on their limitations, such as prescribed placement or limited detection of certain postures or movements [5]. Application of such a sensor does show that monitoring physical activity can already lead to useful insights and can be a tool to improve multidisciplinary care [5]. Therefore, we saw an opportunity to look into the whole chain with a multidisciplinary team, covering practical elements from sensor selection, understanding sensor data with (open-source) algorithms, and real-world interpretations of labeled data over a hospital stay.

The aim of this study was to select and evaluate accelerometers for objective qualification of in-hospital mobilization after cardiac surgery. Based on our previous experience with accelerometers, we chose the investigated accelerometers due to their longer battery-life ability, and practically feasible formats. Objective quantification was defined as determining six static and dynamic patient activities. The hypothesis was that the accelerometer was able to measure patient mobilization activities for every phase of postoperative recovery. First, requirements were formulated, and the available devices reviewed. Second, the best device was selected and applied in a heart surgery ward. Third, the sensor data were automatically interpreted and used to inspect the current mobilization process.

## 2. Materials and Methods

### 2.1. Device Selection

#### 2.1.1. Device Requirements

Technical and clinical requirements were formulated after a literature research and expert consultation with physical therapists, senior researchers in rehabilitation, telemonitoring and eHealth, and a former cardiac surgery patient (Table 1). The considerations of each requirement are depicted in Appendix A.

#### 2.1.2. Review and Selection of Available Devices

Based on the accelerometers we encountered in a broader literature search, eighteen devices were deemed suitable for the requirements analysis.

Alphabetically, the ActiGraph wGT3X-BT (ActiGraph, LLC, Pensacola, FL, USA), Activ8 A8015 (Remedy Distribution Ltd., Watford, UK), activPAL (PAL Technologies Ltd., Glasgow, UK), Alvita Ultimate HJ-325 (Omron Healthcare Inc., Lake Forest, IL, USA), AX3 (Axivity Ltd., Newcastle upon Tyne, UK), MOX1 (Maastricht Instruments B.V., Maastricht, The Netherlands), BCGMCU/SCA11H (Murata Manufacturing Co., Ltd., Nagaokakyo, Japan), Fibion (Fibion Inc., Jyväskylä, Finland), Health Watch (Philips, Drachten, The Netherlands), HeartGuide (Omron Healthcare Inc., Lake Forest, IL, USA), Motiv Ring (Motiv Inc., Seattle, WA, USA), MoveMonitor (McRoberts B.V., The Hague, The Netherlands), SenseWear Pro3 (Micro Star Instruments Co. Ltd., Taipei, Taiwan), VitalPatch (Vital Connect Inc., San Jose, CA, USA), VivoSmart (Garmin B.V., Eemnes, The Netherlands), and Wellsense Vu (Wellsense Inc., Birmingham, MI, USA) were considered. The technical and clinical requirements’ evaluation is shown in Appendix A.

We found three devices to be eligible for use in a clinical setting (ActiGraph, AX3, and MOX1), which met all the technical requirements. The other fifteen devices did not meet the technical requirements. Decisive factors on device selection were based on sample rate with battery life and price. MOX1 (€195) had a battery life less than seven days with a sample rate between 60 and 100 Hz. The ActiGraph (€225) was almost twice as expensive as AX3 (€123). The AX3 was selected for a clinical pilot study, based on the combination of price, access to data, and no need for licensing or purchasing dedicated software.

### 2.2. Classification of Physical Activity Using Accelerometry

A triaxial accelerometer measures the sum of inertial and gravitational acceleration in three directions [23]. It consists of a mass, suspended by a spring in a housing. Tri-axial accelerometers have three single axis accelerometers mounted in a box with sensitive axes in different directions, or as used in the AX3, have a sensor constructed using a single mass [24]. Acceleration is directly proportional to an external force and thus reflects the intensity and frequency of human movement. Accelerometers provide inclination sensing in response to gravity with respect to reference planes when the accelerometers rotate. The resulting inclination data is used to classify body posture orientations. Therefore, the measured accelerometry data can identify postures and daily movements using a classifier. The results can be related to a patient’s functional status [25,26].

The positions and orientations of body segments are stable over time during static activities (e.g., lying, sitting, and standing). Static activities can therefore be identified by body segment orientation with respect to gravitation [27]. Positions and orientations of body segments vary over time during dynamic activities. During dynamic activities (e.g., walking, cycling, and walking stairs), body segments move naturally in a cyclical manner.

For classification of activities, fixed-threshold classification [28,29,30], reference-pattern-based classification [27], pattern-recognition strategies with statistical algorithms [27], conventional or fuzzy logic [31], and artificial neural networks [32,33] were investigated. Machine learning algorithms, including deep learning algorithms, demonstrated high classification accuracy for human physical activity from tri-axial accelerometer data [34,35,36].

The leg and arm were chosen for placement of the sensors, informed by recurring accelerometer placement for human activity recognition [37] and the expected comfort restrictions. Wound location, and expected added value of measuring both the upper and lower body to differentiate lying, sitting, and standing, were also considered. To investigate placement on the arm and leg, and the resulting accelerometer data, a pilot was run with 5 non-clinical subjects wearing 8 sensors. This context-dependent decision process led to the choice of placement of 2 sensors, as seen in Figure 1.

The next step towards classification is choosing an appropriate type of algorithm in combination with the provided (preprocessing of) data. Body acceleration was obtained by passing the acceleration signal through a high-pass filter with a cut-off frequency of 0.8 Hz. The measured data were divided into subsequent segments of 256 samples (2.56 s) based on an average person’s walking cadence. For the selected activities, it was possible to efficiently provide correctly labeled data by giving participants instructions to perform all 6 activities in an ordered calibration session. Based on the provision of a labeled data set and re-purposing existing open-source code for human activity recognition [38], an artificial neural network (ANN) was implemented.

### 2.3. Artificial Neural Network Algorithm Implementation and Validation

The ANN was trained by generating a number of features from the raw data stream. Previously indicated time/frequency-domain and heuristic features were used to this end, with presumed a high discriminatory ability and low computational cost [37]. For the training process, the following features previously described by Bunkheila [38] were included before selection: averaged total (i.e., static and dynamic) acceleration, root mean square of dynamic acceleration, autocorrelation of acceleration (height main peak, height, and position of the second peak), spectral peaks (height and position of the first six peaks), and spectral power (total power of five adjacent predefined frequency bands).

In an attempt to further improve recognition by building on Janidarmian et al. [37], the following features were also included: the median of total acceleration, the standard deviation of dynamic acceleration, and the median absolute deviation and magnitude of both total and dynamic acceleration. For simplicity, only the suggested time-domain features from [37] were included. For both sensors, data from all three axes were used, leading to a total of 160 features. The script used a feed-forward network comprised of one hidden layer with eighteen hidden nodes.

A neighbourhood component analysis (NCA) was applied as a feature selection technique. Only features with a feature weight ≥ 0.1 were kept using MATLAB’s feature selection NCA implementation (fsnca [39]), where the default solver type was SGD with an accompanying maximum of 30 iterations. This resulted in a feature selection of 59 features (see Appendix A). The sample size was reduced fitting the class with the lowest number of samples and then randomly selected. Static activities were performed more often than dynamic activities, to a ratio of 3 to 1. The results will be discussed in Section 3.1.2.

A standard K-fold (K = 10) and Leave-One-Subject-Out (LOO) cross-validation was used to determine the performance on the trained data. Where the latter is a better representation to investigate inter-subject differences, the former is a standard cross-validation, leading to comparative results but is susceptible to intra-subject data contamination. The main purpose of the study is alignment towards clinical implementation. This validation was thus not intended to be performance-driven but instead as verification for objective measurement of mobilization.

### 2.4. Clinical Application at a Cardio-Thoracic Surgery Ward

#### 2.4.1. Study Design and Population

The MOV_E_M_E_NTT study (Mobilization Observation and Validation with acceleroMetry at thoraxceNTrum Twente) is a prospective single-center cohort study, reported as per the STROBE recommendations for observational research [40]. Adult patients undergoing cardiac surgery (CABG, valve surgery and combined CABG and valve surgery) between 2 June and 31 July 2020 were recruited at Thoraxcentrum Twente, Medisch Spectrum Twente, a tertiary nonacademic teaching hospital in Enschede, the Netherlands. Patients did not make additional hospital visits, and therefore conformed to hospital measures during the coronavirus pandemic.

Patients with a Katz Index of Independence in Activities of Daily Living functioning (KATZ-ADL) score larger than 2 before surgery (i.e., the patient is not independent in daily life mobilization) [41], patients with an ICU stay exceeding 72 h, patients with postoperative cerebrovascular accidents, and patients mentally incompetent (including postoperative delirium) were excluded from the study. The sample size was determined to be 20 patients for sufficient evaluation of the accelerometer. This is larger than the median of 10 (1–36) patients recently reported in a large survey on accelerometer pattern recognition studies [35].

The study ended at discharge from the hospital and thus no follow-up was obtained. This study was exempted from the Medical Research Involving Human Subjects Act by the Medical Ethics Committee Twente (METC Twente: K20-14) and was approved on 31 March 2020 by the local institutional review board.

Patients were recruited on hospital admission one day before cardiac surgery. After signing informed consent, two AX3 accelerometers were attached lateroproximal on the right upper arm and anterodistal on the right upper leg (Figure 1). Hereafter, standardized exercises were performed, labelling measurements for lying, sitting, standing, walking, cycling, and walking the stairs (see Appendix A for the protocol); then the sensors were removed.

After surgery and ICU discharge, patients were transferred to the cardio-thoracic surgery ward. The sensors were attached, fixed with Tegaderm™ patches, and the measurements started, continuing 24/7. Measurements ended after 7 days or at earlier hospital discharge.

All patients received physical therapy twice a day until the 3rd postoperative day and then once a day. A standardized program starting from ICU discharge included:Day 1: Breathing exercises, coughing techniques, control mobilization of the upper and lower extremities, and transfer from bed to chair with assistance;Day 2: Exercises as on Day 1. Self-transfer from bed to chair with or without assistance, and ambulation with assistance for 20 m at the surgical ward;Day 3: Exercises as on Day 2. Ambulation with an increase in distance (±15 m) and frequency (3 times), cycling for 5–10 min at 0–10 Watt, depending on the hemodynamic stability;Day 4: Exercises as on Day 3. Walking the stairs (1 floor) with assistance, information on home mobilization, and an increase cycling duration (5–10 min) and power (5–10 Watt).

Each patient-specific exercise program was based on the evaluation findings, comorbidities, and patient goals. Patients were stimulated to mobilize outside these sessions by nurses and other caregivers.

#### 2.4.2. Variables and Outcomes

The primary endpoint of this study was the daily duration of each activity performed between 7 a.m. and 11 p.m. in minutes. The secondary endpoints were the length of ICU stay (nights) and surgical ward stay (nights). Routine baseline characteristics were determined based on the body mass index (BMI, kg/m^2^) and European System for Cardiac Operative Risk Evaluation II (EuroSCORE II) definitions [42].

#### 2.4.3. Statistical Analysis

Continuous variables were tested for normality with visual inspection of histograms and skewness/kurtosis measures. A *p*-value of less than 0.05 was set as statistically significant. Variables were analyzed with unpaired *t*-tests or Mann–Whitney U tests for continuous variables and for categorical variables with a Fisher Exact test.

Accuracy is defined as the ratio (%) of correct predictions to the total input samples. Precision, or positive predictive value, is the ratio (%) of correct classified samples of all samples belonging to the respective class. Recall, or the true positive rate, is the ratio (%) of correct classifications of the samples that should have been identified in that class.

A prespecified gender subgroup analysis, as encouraged by the Institute of Medicine [43], was performed. After inclusion of 20 patients, only 15% was female. As this was lower than anticipated, the ethical review board approved the amended protocol to include up to 10 women (approved date: 9 July 2020). A linear mixed model with maximum likelihood was used to determine differences in activity time over time (day-to-day minutes and percentages), and the differences between male and female patients. Results are reported as the mean ± SD, median (with the 25th to 75th percentiles), or mean (min–max). Frequencies are reported as *n* (%).

## 3. Results

### 3.1. Clinical Study

#### 3.1.1. Patient Characteristics

Out of 75 patients selected for eligibility, in total 44 patients were excluded based on inability to perform the labelling measurement (*n* = 18), patient transfer from ICU to surgical ward planned during weekend days (*n* = 9), no available sensors (*n* = 8), patients referred from other hospitals (*n* = 6), or patients unwilling to participate (*n* = 3). Thus, 31 patients were included for the preoperative labelling measurements. Two patients had an ICU stay longer than 72 h and were excluded from the analysis. The CONSORT study flow diagram [44] is shown in Appendix A.

In this study, 22 male and 7 female patients were included, with no differences in their baseline (Table 2) or periprocedural (Table 3) characteristics, except for patient length (*p* < 0.001) and body surface area (*p* = 0.003). The majority of patients received isolated CABG (52%), 31% received valve surgery, and 17% CABG with concomitant valve surgery. Almost 47% of the performed CABG were handled off-pump, thus without cardiopulmonary bypass (7 out of 15 patients).

The median ICU stay was 1 (1 to 2) night with no difference between male (1 (1 to 2)) and female (2 (1 to 2)) patients (U = 49, *p* = 0.13). Median surgical ward stay was 5 (3 to 6) nights, without a significant difference between male (5 (3 to 6)), and female (5 (3 to 7)) patients (U = 70, *p* = 0.73).

Thoracic drains were removed after 44 (39 to 47) hours after surgery. No perioperative myocardial infarction, prolonged intubation, readmission to the ICU, cerebral vascular accident, kidney failure, or gastrointestinal ischemia were observed. One pulmonary infection was observed in the male subgroup (4.5%), one rethoracotomy in both groups (male 4.5%, female 14%, *p* = 0.43) and one deep sternal wound infection occurred in the male group (4.5%). No patients died during hospital stay.

#### 3.1.2. Preoperative Labelling Measurements

Labelled data from 31 participants was used for neural network training. Patients were asked to subsequently stand, sit on a chair, and lie on a bench on their back, right, and left side. Afterwards, patients walked back and forth on the corridor for 10 m, cycled on an exercise bike, and took one flight of stairs up and down. Each posture and activity had a duration of 30 s. On average, 11 m of labelled data were used for each patient. Segments of measured data were manually labelled by a researcher and plotted with MATLAB (see Appendix A). With visual inspection of the plots, the data samples were labelled as one of the six activities. For training purposes, a 50% overlap was used. No overlap was used for the actual postoperative activity classification.

The algorithm showed a 98% overall accuracy, as determined by a 10-fold cross-validation and 96% for LOO validation using 160 or 59 features. All activities exceeded 90% on recall and precision (Table 4).

The overall classification accuracy of the neural network, and that for male and female patients, started and remained constant at 95 to 96% throughout the study (Appendix A). Both the recall and precision of the neural network was at least 78% (lying in bed) and increased over 90% throughout the study for all individual activities (Appendix A).

#### 3.1.3. Postoperative Mobilization Activities

Physical activities were obtained from 29 patients for 4 (2 to 5) days. One patient (3.4%) reported inconveniences regarding placement of the accelerometers. Patients spent 41 (20 to 62) min less time in bed for each consecutive hospital day, as determined by the mixed-model analysis (*p* < 0.001). For an overview of the relative time spent for all activities per hospital day, see Figure 2.

For all individual activities, see Table 5 for the fixed effects analysis and Figure 3 for the absolute and relative time spent for each activity per postoperative day. Thus, the model can be described by Equation (1):(1)$$Y_{j}=(\beta_{0}+\beta_{1})+\beta_{2}Time_{j}$$
where *Y_j_* is the patient activity as a function of the *j*th day, *β*_0_ is the mean estimate of Day 1 (intercept), *β*_1_ the increase of male over female patients, *β*_2_ the slope of the time component, and Time is the number of days after Day 1 as a function of the *j*th day.

The relative percentages of each activity changed during hospital stay (*p* = 0.01 for cycling and *p* < 0.001 for all other activities). For both absolute activity duration and relative percentage, no significant differences between men and women were observed (Table 5).

Non-sedentary activities, such as standing, walking, and walking the stairs, increased during hospital stay (all *p* < 0.004), but not cycling (*p* = 0.07). From Day 3, the time spent cycling was stable (Figure 3i). Accelerometers are not able to measure an increase in resistance for exercise bikes. Therefore, no improvement could be measured for cycling during this study.

Anthropometric parameters, such as patient length, weight, and body surface area, were not in the original protocol, but patient length and body surface area were significantly different between male and female patients. Patient length and body surface area were added as a single covariate to the linear mixed models for all primary endpoints. Length was a significant covariate in the model for lying in bed with only time (days after ICU discharge), F (1, 23.892) = 6.089, *p* = 0.02. Body surface area was also significant in a model for lying in bed, F (1, 46.359) = 4.173, *p* = 0.047.

Some patient movements were classified as cycling or walking the stairs on Day 1. According to the protocol and patient files, none of these activities were reported on Day 1. Misclassifications of the algorithm might occur when patients transfer between activities. As these transfers are of short duration and is similar for all patients for all postoperative days, this should have no effect on our interpretation of the results.

## 4. Discussion

The aim of this study was to select and evaluate accelerometers for objective qualification of in-hospital mobilization after cardiac surgery with six static and dynamic patient activities. This study was designed in a practical manner, to be relevant for healthcare professionals, researchers, and patients.

### 4.1. Device Selection

After the literature review and expert consultation, three sensors were deemed suitable for objective qualification of in-hospital mobilization after cardiac surgery. The final choice for the AX3 sensor was mainly based on price, battery life, openly accessible sensor data, and not purchasing additional software. Many stakeholders were involved, including rehabilitation and sensor researchers, physical therapists, a cardiac surgeon, and a patient. In our current study, only accelerometer data were considered; future studies might consider different types of data to further improve classification [35].

### 4.2. Clinical Evaluation

The aforementioned approach to measure mobilization activities appeared feasible. There were no technical problems with the AX3 devices as all data was usable for the analysis, and the Tegaderm™ patches did not unbind prematurely from the skin. One patient experienced some inconvenience from wearing the sensors. This study combined two accelerometers placed at both the upper and lower extremity. To the best of our knowledge, no research has been conducted with this approach to objectify in-hospital mobilization after cardiac surgery. All six activities were performed by patients and were recorded with the accelerometer.

In a previous study, an increase in the metabolic equivalent of tasks (METs) ≥3 and step count were observed after cardiac surgery in 83 patients receiving low-intensity physical therapy [12]. Light activities (METs < 3) increased from 3 ± 6 min on the 1st day after ICU discharge to 22 ± 24 min at Day 4. In our study, both light activities (standing and walking), and activities with moderate intensity (cycling and walking the stairs) were more frequently performed and earlier during postoperative stay. Light activities in our study were performed for 1 (1 to 5) min on the 1st day and increased to 55 (41 to 100) min on the 4th day. The difference can be explained by having standard physical therapy for all patients and using more active mobilization activities in our setting.

Previous studies reported higher functional levels and faster mobilization for male patients [12,13]. Men might have a more competitive attitude to mobilization or overestimate self-reported activities. Women also might be more hesitant to mobilize after surgery. In this study, no effect of sex was observed for any of the activities observed.

Some of the baseline and perioperative and postoperative variables might affect mobilization activities. The original study protocol did not include patient length and body surface area. It turned out, as expected, that these parameters are significantly different between male and female patients. Both factors were significant covariates in a linear mixed model for quantifying lying in bed only. Both parameters remained significant in a model with sex as fixed factor. However, lying in bed was still not significant between male and female patients (*p* = 0.08). Future studies should include anthropometric parameters, i.e., patient length, weight, BMI, and body surface area, to correct for differences in the baseline characteristics between male and female patients.

To generate hypotheses for future studies, all parameters from Table 2 and Table 3 were added to the linear mixed model as a fixed factor or covariate. Although *p*-values < 0.05 can be considered as significant, with these multiple comparisons, type I errors are likely to occur. Only length and body surface area were significant covariates for the static activities. For the dynamic activities, only previous neurological dysfunction had an impact on two activities: walking (both absolute and relative) and walking the stairs (both absolute and relative). The results from the mixed models are added as Appendix A.

An accelerometer for measuring inpatient mobilization was tested for 29 patients. A recent survey showed that previous studies included a median of 10 (1–36) patients [35]. The neural network precision and recall per activity for this study exceeded 78%, as determined with LOO validation over *n* number patients, even from *n* = 2. After *n* = 10, both the precision and recall for all individual activities exceeded 90%, and remained stable (Appendix A). Nonetheless, the difference between K-fold (K = 10) and LOO cross-validation, especially looking at the range and difference in training size, might warrant careful inspection. This calls for future work to investigate an analysis on unseen subjects. The sample size with the calibration step included is deemed sufficient to draw conclusions on the neural network performance. No differences in neural network performance between male and female patients were observed.

The study was designed for measuring, with two accelerometers, all six activities after cardiac surgery. As a sensitivity analysis, the accuracy of only one accelerometer was also calculated. With the upper arm sensor only, an accuracy of 80% (77–83) was obtained with K-fold cross validation, and 74% (17–95) with LOO validation. For the upper leg, the neural network scored the activities accurate for 95% (94–95) and 92% (76–100), respectively. For future studies, only one accelerometer on the upper leg should be considered for measuring the postoperative mobilization activities after cardiac surgery. It is likely that our approach is suitable for classification of mobilization for other types of surgery or other patient groups, as long as sensor placement does not affect the wound area. Furthermore, these in-hospital sensor data might prove beneficial as a baseline measurement for the intake of outpatients for cardiac rehabilitation.

### 4.3. Limitations

Preoperative labeling might have introduced misclassifications as all patients walked without support. Patients in their early postoperative stay walk slower and sometimes with a walker or physical therapist support. This might have overestimated standing and underestimated walking. Postoperative activity forms were designed to estimate mobilization activities from a professional perspective, and compare these with the classifications. These forms were too generic to detect misclassifications. However, the data retrieved from the forms were consistent with the actual classifications.

For time-efficiency reasons, only the reduced feature set was used to classify the activities after surgery, although in retrospect a more thorough NCA threshold choice for feature selection is advised for future work [45]. The cutoff ≥0.1 was deemed appropriate based on MATLAB’s fcnca function [39], and visual inspection of the histograms. No sensitivity analysis was performed on setting the cutoff for features larger than zero, but which is recommended for future research.

Patients were discharged from ICU staring at 10:30 a.m. and therefore the 1st day measurements missed some hours in the absolute data analysis. Indeed, low absolute time spent on lying in bed and sitting (Figure 3a,c) was observed on Day 1. Using proportional time (Figure 3b,d) might reflect the activities better on days with incomplete data collection. Using both parameters should be used for future studies as well.

Small positional changes of the sensors have strong effects on the sensor output [46]. Before our clinical pilot study started, four positions were considered on both the arm and leg. The best positions were identified in a calibration study in four healthy subjects. Potential sensor position changes during the study were not identified. To increase reliability, a user-friendly position of the sensor, no clipping of true accelerations, and a tight body fit were recommended [46]. The sensor placement in our study on the upper leg might warrant further inspection from a biomechanical standpoint due to possible inclusion of soft tissue (e.g., muscle movement). Importantly, this sensor in combination with placement on the upper arm did not hamper free movement. Furthermore, 97% of the included patients did not report inconvenience. Sensors for the preoperative calibration measurements and postoperative quantification were attached by one researcher according to prespecified landmarks. With this approach, a tight fit and similar positions were achieved.

The clinical pilot was designed to classify in-hospital mobilization activities. Future research should investigate how this information can be used to improve rehabilitation. Physical therapists and nurses can tailor their strategies with reliable information on performed activities. With persuasive designs and serious gaming, patients could be motivated to achieve higher levels of functional independence. New technical and clinical requirements might be formulated, because direct feedback is different from analysis after hospital discharge. The next step is to provide feedback to patients and healthcare professionals, to speed up recovery.

## 5. Conclusions

Commercially available devices for in-hospital mobilization after cardiac surgery were screened for suitability in a literature review and one device selected for evaluation. Static and dynamic activities, e.g., lying, sitting, standing, walking, cycling, and walking the stairs, were classified with an AX3 accelerometer for postoperative patients at a cardiac surgery ward. Patients spent 41 (20 to 62) min less time in bed for each consecutive hospital day (*p* < 0.001) based on the sensor data. Non-sedentary activities, such as standing, walking, and walking the stairs, increased during the hospital stay (all *p* < 0.004), but not cycling (*p* = 0.07). No differences between male and female patients were observed for all endpoints.

The approach presented in this study deems applicable for monitoring postoperative recovery of patients in similar contexts.

## Figures and Tables

**Figure 1 sensors-21-01979-f001:**
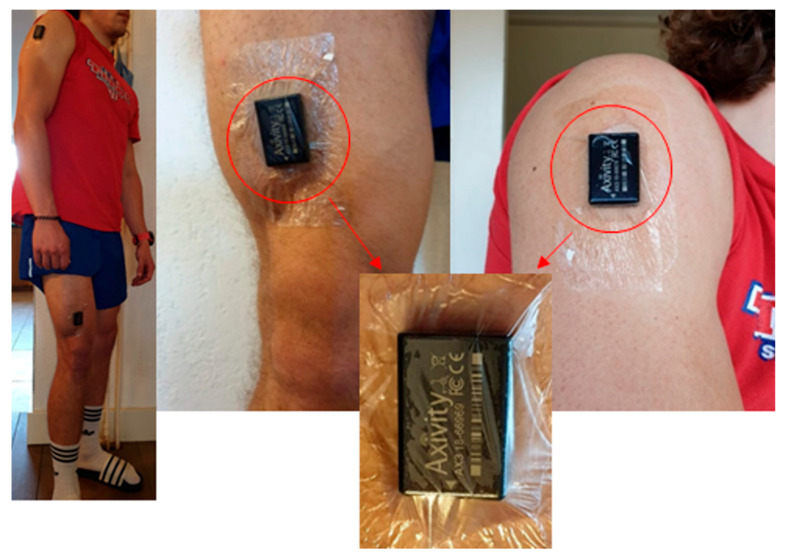
The AX3 accelerometers were placed lateroproximal on the right upper arm and anterodistal on the right upper leg. Permission was obtained for use of this photograph for publication.

**Figure 2 sensors-21-01979-f002:**
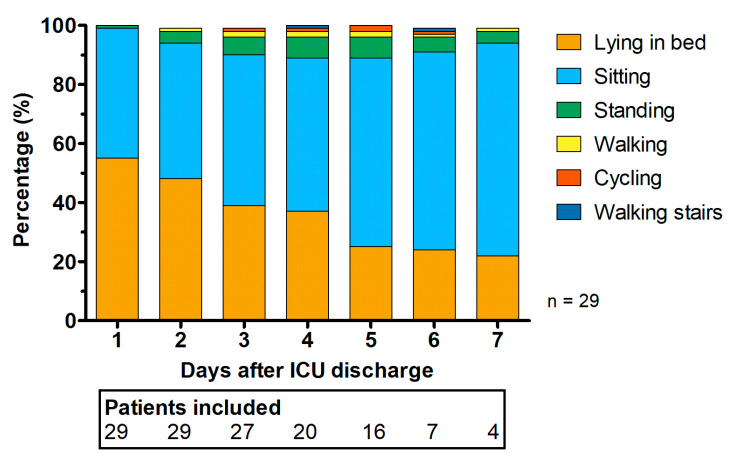
Relative time spent per activity per postoperative day at the cardio-thoracic surgery ward between 7 a.m. and 11 p.m. A decrease in lying in bed (orange) and an increase in sitting (sky blue) were observed during the hospital stays. The number of included patients decreased because of the varying length of stay and discharge from the hospital.

**Figure 3 sensors-21-01979-f003:**
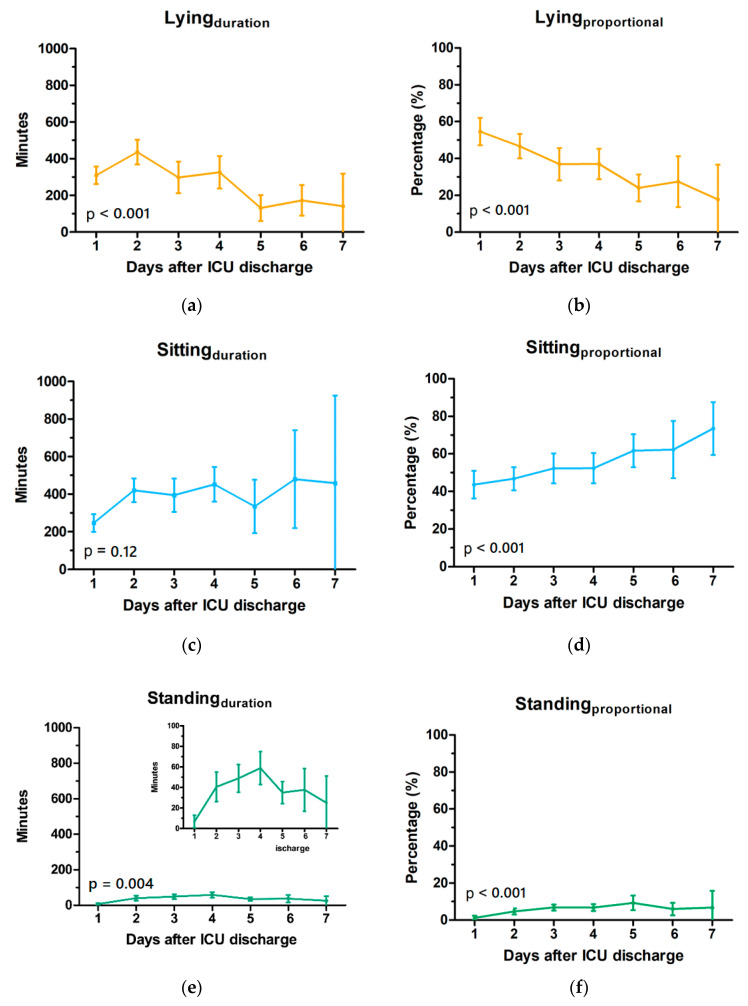
Patient activities at the cardio-thoracic surgery ward as labelled with the average time spent on each activity between 7 a.m. and 11 p.m. on the left panels, and the relative time as percentage on the right panels: (**a**,**b**) lying in bed; (**c**,**d**) sitting in a chair; (**e**,**f**) standing; (**g**,**h**) walking; (**i**,**j**) cycling; (**k**,**l**) walking the stairs. Lines represent the mean with 95% confidence interval and the *p*-values depict the effect of time (days) on each activity. Fixed axes are used for comparison with additional inlays to show details.

**Table 1 sensors-21-01979-t001:** Requirements for a device to classify in-patient physical activities after cardiac surgery, ranked from highest to lowest importance.

	Technical Requirements		Clinical Requirements
1.	Classification of various activities Static: Lying, sitting, standing Dynamic: Walking, cycling, walking the stairs	1.	No burden during the hospital stay and no hindrance in mobilization
2.	Sampling rate between 60 and 100 Hz	2.	Minimal soft tissue deformation
3.	Battery life for measuring up to 7 consecutive days	3.	Device placement away from wound area
4.	Local data storage up to 7 consecutive days	4.	Meeting regulations regarding patient privacy and integrity
5.	Data extraction from the device	5.	No impediment in daily nursing care
		6.	Device is CE certified
		7.	Total costs for clinical pilot study should not exceed €3000
		8.	Data analysis is attainable with supplied software and/or is available with a University license

**Table 2 sensors-21-01979-t002:** Baseline characteristics of the included patients.

Variable	Total (*n* = 29)	Male (*n* = 22)	Female (*n* = 7)	*p* Value
Age, years	70 (64 to 74)	72 (64 to 74)	68 (56 to 75)	0.52
Length, cm	176 (170 to 180)	179 (172 to 183)	167 (164 to 169)	<0.001
Weight, kg	84 ± 12	86 ± 11	79 ± 12	0.16
Body mass index, kg/m^2^	27 (25 to 30)	26 (25 to 293)	29 (24 to 36)	0.44
Body surface area (Du Bois), m^2^	2.00 ± 0.15	2.04 ± 0.13	1.86 ± 0.13	0.003
Diabetes, *n* (%)	6 (21)	3 (14)	3 (43)	0.13
Multivessel disease, *n* (%)	13 (45)	11 (50)	2 (29)	0.41
Recent myocardial infarction, *n* (%)	2 (6.9)	2 (9.1)	0 (0.0)	1.0
LVEF, *n* (%)				0.23
Poor, <30%	4 (14)	4 (18)	0 (0.0)	
Moderate, 30–50%	12 (41)	7 (32)	5 (71)	
Good, >50%	13 (45)	11 (50)	2 (29)	
COPD, *n* (%)	1 (3.4)	1 (4.5)	0 (0.0)	1.0
Extracardiac arteriopathy, *n* (%)	3 (10)	3 (14)	0 (0.0)	0.56
Neurological dysfunction, *n* (%)	3 (10)	2 (9.1)	1 (14)	1.0
Previous cardiac surgery, *n* (%)	0 (0.0)	0 (0.0)	0 (0.0)	1.0
NYHA class, *n* (%)				0.54
I	12 (41)	8 (36)	4 (57)	
II	7 (24)	5 (23)	2 (29)	
III	10 (35)	9 (41)	1 (14)	
IV	0 (0.0)	0 (0.0)	0 (0.0)	
Urgency, *n* (%)				0.29
Elective	23 (79)	16 (73)	7 (100)	
Urgent	6 (21)	6 (27)	0 (0.0)	
Emergent/Salvage	0 (0.0)	0 (0.0)	0 (0.0)	
Preoperative hemoglobin, mmol/L	8.7 ± 1.2	8.8 ± 1.2	8.2 ± 0.98	0.22
Preoperative aspirin	18 (62)	15 (68)	3 (43)	0.38
Preoperative anticoagulant agents	1 (3.4)	1 (4.5)	0 (0.0)	1.0
EuroSCORE I, logistic (%)	3.52 (1.47 to 6.09)	3.76 (1.49 to 6.04)	3.50 (1.22 to 6.31)	0.82
EuroSCORE II, logistic (%)	1.34 (0.89 to 2.01)	1.44 (0.89 to 2.36)	0.95 (0.69 to 1.53)	0.30

Values are expressed as *n* (%), mean ± standard deviation, or median (25th to 75th percentiles). 1 mmol/L hemoglobin = 1.61 g/dL hemoglobin. COPD = Chronic obstructive pulmonary disease; EuroSCORE = European system for cardiac operative risk evaluation; LVEF = Left ventricular ejection fraction; NYHA = New York Health Association.

**Table 3 sensors-21-01979-t003:** Periprocedural and postoperative characteristics of the included patients.

Variable	Total (*n* = 29)	Male (*n* = 22)	Female (*n* = 7)	*p* Value
Type of surgery, *n* (%)				0.84
CABG	15 (52)	12 (55)	3 (43)	
Valve surgery	9 (31)	6 (27)	3 (43)	
CABG + valve surgery	5 (17)	4 (18)	1 (14)	
Cardiopulmonary bypass, *n* (%)	22 (76)	16 (73)	6 (86)	0.65
ICU stay, nights	1 (1 to 2)	1 (1 to 2)	2 (1 to 2)	0.13
Surgical ward stay, nights	5 (3 to 6)	5 (3 to 6)	5 (3 to 7)	0.73
Thoracic drains in situ, hours until removal	44 (39 to 47)	44 (39 to 47)	43 (39 to 77)	0.74
Discharge to, *n* (%)				1.0
Home	22 (76)	17 (77)	5 (71)	
Referring hospital	7 (24)	5 (22)	2 (29)	

Values are expressed as *n* (%) or median (25th to 75th percentiles). CABG = Coronary artery bypass grafting; ICU = Intensive care unit.

**Table 4 sensors-21-01979-t004:** Preoperative labelling of activities with K-fold cross validation (K = 10) and LOO validation.

Patient Activity	K-Fold (K = 10)		LOO (*n* = 31)	
	160 Features	59 Features	160 Features	59 Features
**Stationary Activities**				
*Lying in bed*, *%*				
Recall	97 (92–99)	99 (96–99)	94 (59–100)	94 (66–100)
Precision	98 (95–99)	98 (96–100)	93 (71–100)	91 (53–100)
*Sitting*, *%*				
Recall	98 (95–100)	98 (98–100)	93 (12–100)	92 (12–100)
Precision	98 (93–99)	99 (97–99)	94 (71–100)	94 (75–100)
*Standing*, *%*				
Recall	99 (98–100)	99 (98–100)	99 (91–100)	99 (91–100)
Precision	100 (100–100)	99 (98–100)	99 (85–100)	99 (87–100)
**Dynamic activities**				
*Walking*, *%*				
Recall	98 (96–100)	95 (96–99)	98 (82–100)	99 (91–100)
Precision	94 (91–96)	98 (92–98)	94 (82–100)	99 (87–100)
*Cycling*, *%*				
Recall	100 (100–100)	100 (99–100)	99 (86–100)	97 (77–100)
Precision	100 (100–100)	100 (99–100)	99 (76–100)	93 (82–100)
*Walking the stairs*, *%*				
Recall	94 (90–96)	98 (92–99)	93 (42–100)	93 (42–100)
Precision	98 (97–99)	96 (96–100)	97 (85–100)	97 (85–100)
**Accuracy**, **%**	98 (97–98)	98 (98–99)	96 (85–100)	96 (85–100)

Data are median (min–max).

**Table 5 sensors-21-01979-t005:** Patient activities during their hospital stay between 7 a.m. and 11 p.m., as measured with the accelerometer, and the fixed effects analysis (*n* = 29).

Patient Activity	Mean Estimate Day 1 (95% CI) (Intercept)	Increase per Day (95% CI) (Time)	Overall Increase for Men Compared to Women (95% CI)
**Stationary Activities**			
*Lying in bed*			
Minutes per day	413 (340 to 487)	−41 (−62 to −20), *p* < 0.001	11 (−78 to 100), *p* = 0.81
Percentage per day	60% (52 to 68)	−6.5% (−8.7 to −4.2), *p* < 0.001	0.05% (−11 to 11), *p* = 0.99
*Sitting*			
Minutes per day	294 (208 to 379)	19 (−5.3 to 44), *p* = 0.12	−72 (−175 to 31), *p* = 0.17
Percentage per day	40% (32 to 48)	3.8% (1.7 to 6.0), *p* < 0.001	−2.1% (−13 to 9.0), *p* = 0.70
*Standing*			
Minutes per day	14 (−0.1 to 28)	5.7 (1.8 to 9.5), *p* = 0.004	−2.0 (−21 to 17), *p* = 0.83
Percentage per day	−0.57% (−2.7 to 1.5)	2.1% (1.6 to 2.7), *p* < 0.001	1.4% (−2.0 to 4.7), *p* = 0.41
**Dynamic activities**			
*Walking*			
Minutes per day	−0.39 (−2.6 to 1.8)	2.0 (1.5 to 2.6), *p* < 0.001	1.1 (−2.4 to 4.7), *p* = 0.53
Percentage per day	−0.12% (−0.91 to 0.67)	0.55% (0.36 to 0.74), *p* < 0.001	0.81% (−0.48 to 2.1), *p* = 0.21
*Cycling*			
Minutes per day	1.4 (−3.9 to 6.8)	1.4 (−0.13 to 2.9), *p* = 0.07	4.2 (−2.4 to 11), *p* = 0.21
Percentage per day	−0.08% (−0.74 to 0.58)	0.24% (0.06 to 0.43), *p* = 0.01	0.51% (−0.34 to 1.4), *p* = 0.23
*Walking the stairs*			
Minutes per day	0.42 (−1.0 to 1.9)	0.68 (0.27 to 1.1), *p* = 0.001	0.64 (−1.3 to 2.5), *p* = 0.50
Percentage per day	−0.15% (−0.40 to 0.09)	0.17% (0.11 to 0.24), *p* < 0.001	0.13% (−0.20 to 0.46), *p* = 0.43

Values are expressed as the estimate effect (95% confidence interval with the lower and upper boundaries).

## Data Availability

Data are available from the authors upon request.

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
