# Peer review of "Objective Quantification of In-Hospital Patient Mobilization after Cardiac Surgery Using Accelerometers: Selection, Use, and Analysis"

_sensors, 2021, doi:10.3390/s21061979_

Round 1

Reviewer 1 Report

The authors described an interesting study to select and evaluate accelerometer for objective qualification of in-hospital mobilization after cardiac surgery. They explored 18 devices and found three sensors were suitable. They finally selected AX3 and used some extracted features along with ANN to classify 6 different activities.

The manuscript can be improved by addressing the following comments:

Major comments:

  1. What kind of features did the authors use? They mentioned only time/frequency domain features but the feature names/extraction methods are missing. If all the features are beyond description, then they need to mention few feature names (at least) in the manuscript to give the reader some insights.
  2. How the NCA was applied for feature selection? How the threshold for NCA weights was selected? NCA gives weights for all the features. So, the threshold (or regularizer parameter) is an important choice. Authors can see this paper (10.1109/EMBC44109.2020.9176239) where NCA based feature selection and DNN are used for activity classification from accelerometer data.
  3. 11 minutes of data were used for the neural network training. The authors probably divided the data into some segments/windows for calculating the features, but it is not clear. They need to clarify the window/segment length (if used).
  4. How about the sample distribution across the 31 subjects (for ANN model training)? For example, how many total samples are used, what is the sample number for each class (each activity) etc. are needed to be mentioned.

Minor comments:

  1. Details description of the ANN can be added. How many iterations/epochs are used for training and how was the accuracy/loss varying with epochs/iterations?
  2. No description of the total features and subset of features are given. How many features were chosen before and after the NCA selection?
  3. The authors can revise the conclusion section to better summarize the important findings of this study.
  4. The manuscript needs to be revised for grammatical errors and/or typos. In the abstract: “lying in bed, sitting in a chair, walking, cycling on an exercise 24 bike and walking the stairs”. Probably there will be 6 activities.

Reviewer 2 Report

To: Editorial Board

Sensors

Title: “Objective quantification of in-hospital patient mobilization after cardiac surgery using accelerometers: Selection, Use, and Analysis”

Dear Editor,

I read this manuscript and I think that:

  • The results section of the abstract should include more numerical data. Please update this section.
  • Table 1 should be updated by considering all the characteristics of the study population: anthropometric, laboratory, clinical, and pharmacological history of the patients should be considered.
  • A multivariate regression analysis should be considered in order to evaluate the role of confounding factors on final results.
  • the role of care manager should be considered for the final application and results of the paper. Please discuss such a point in relation to the paper from Ciccone MM et al. Vasc Health Risk Manag. 2010 May 6;6:297-305.

Round 2

Reviewer 1 Report

The authors have addressed all my comments and I don't have any further concerns.

Reviewer 2 Report

the authors well addressed my previous comments. the paper improved very much.